# Role of Extracellular High-Mobility Group Box-1 as a Therapeutic Target of Gastric Cancer

**DOI:** 10.3390/ijms23063264

**Published:** 2022-03-17

**Authors:** Wataru Takaki, Hirotaka Konishi, Daiki Matsubara, Katsutoshi Shoda, Tomohiro Arita, Satoshi Kataoka, Jun Shibamoto, Hirotaka Furuke, Kazuya Takabatake, Hiroki Shimizu, Shuhei Komatsu, Atsushi Shiozaki, Takeshi Kubota, Kazuma Okamoto, Eigo Otsuji

**Affiliations:** 1Division of Digestive Surgery, Department of Surgery, Kyoto Prefectural University of Medicine, Kyoto 602-8566, Japan; w-takaki@koto.kpu-m.ac.jp (W.T.); daiki425@koto.kpu-m.ac.jp (D.M.); kshoda@yamanashi.ac.jp (K.S.); t-art@koto.kpu-m.ac.jp (T.A.); kataokas@koto.kpu-m.ac.jp (S.K.); shibamo@koto.kpu-m.ac.jp (J.S.); hfuruke@koto.kpu-m.ac.jp (H.F.); batake@koto.kpu-m.ac.jp (K.T.); hiro0810@koto.kpu-m.ac.jp (H.S.); skomatsu@koto.kpu-m.ac.jp (S.K.); shiozaki@koto.kpu-m.ac.jp (A.S.); tkubot@koto.kpu-m.ac.jp (T.K.); kazuma@koto.kpu-m.ac.jp (K.O.); otsuji@koto.kpu-m.ac.jp (E.O.); 2First Department of Surgery, Faculty of Medicine, University of Yamanashi, Kofu 400-8510, Japan

**Keywords:** high-mobility group box-1, recombinant human soluble thrombomodulin, gastric cancer

## Abstract

Background: High-mobility group box-1 (HMGB1) is involved in the tumorigenesis and metastasis of various cancers. The present study investigated the roles of extracellular HMGB1 in the progression of gastric cancer (GC) and the therapeutic effects of recombinant human soluble thrombomodulin (rTM) targeting HMGB1. Methods: The effects of extracellular HMGB1 and rTM on GC cells were assessed using proliferation and Transwell assays. Their effects on local tumor growth and metastasis were evaluated using subcutaneous tumor and liver metastasis mouse models, respectively. Plasma HMGB1 concentrations in GC patients were measured using ELISA. The relationships between plasma HMGB1 concentrations and the prognosis and clinicopathological factors of patients were also investigated. Results: GC proliferation, migration, and invasion abilities were promoted by increases in extracellular HMGB1 concentrations and alleviated by rTM. In the subcutaneous tumor model, local tumor growth was promoted by the addition of rhHMGB1 and alleviated by rTM. Similar changes occurred in the liver metastasis model. Recurrence-free survival (*p* < 0.01) and overall survival (*p* = 0.01) were significantly worse in patients with high plasma HMGB1 concentrations. Conclusion: Plasma HMGB1 concentrations are a prognostic marker in GC patients. Extracellular HMGB1 promotes cancer progression and has potential as a novel treatment target in GC cells for rTM.

## 1. Introduction

Gastric cancer (GC) is the sixth most common malignant tumor worldwide and the third leading cause of cancer-related death due to its malignancy [1]. The standard treatment for resectable advanced GC in the Japanese gastric cancer treatment guidelines is radical gastrectomy with lymphadenectomy followed by adjuvant chemotherapy [2]. Although surgical techniques and chemotherapies, for example, the indication of immune-checkpoint inhibitors, are improving, the prognosis of patients with advanced cancer is still poor and metastatic recurrence is more likely to occur as the cancer stage increases [3,4]. Pre-operative micrometastasis has been suggested as a cause of recurrence; however, the mechanisms that promote micrometastasis have not yet been elucidated, and the development of novel therapeutic strategies that target these factors is awaited.

Regarding carcinogenesis of gastric cancer, the protease activated receptors (PARs), seven transmembrane-spanning domain G protein-coupled receptors, are in the spotlight. They activate multiple intracellular signals through the interaction of both environmental and genetic factors, leading to the proliferation and survival of GC cells [5]. Endothelial protein C receptor (EPCR) and PARs play important roles in both cancer growth and dissemination, and, moreover, the interactions between the two receptors are essential for tumor progression [6,7]. EPCR and PAR-1 share an activated protein C (APC)-dependent pathway, and this pathway may have an essential function in cancer progression, including GC [8,9]. PARs are also expressed in the epithelium of the gastrointestinal tract, suggesting that microbiota may act on this pathway and exert a carcinogenic effect [10]. Especially recently, the interaction between tumor progression and blood coagulation has been focused on. APC exerts normal anticoagulant and fibrinolytic functions, as well as complexing with EPCR to cleave PAR-1 and exert anti-tumor effects [11]. Although recombinant human APC may work as an anti-tumor agent, its short half-life and the risk of hemorrhagic complications make its practical application difficult [12]. For this reason, we focused on high-mobility group box-1 (HMGB1) as another molecule associated with these pathways.

HMGB1 is a highly conserved nuclear protein that is present in almost all biological organisms and acts as a chromatin-binding factor that facilitates the assembly of DNA-binding proteins into their specific binding targets [13,14]. In addition, following exposure to cellular stress and damage, HMGB1 translocates into the cytoplasm and is secreted extracellularly to function as an extracellular signaling molecule during inflammation and tumorigenesis [13,14,15,16,17,18]. Previous studies reported that the overexpression of HMGB1 is involved in the development and metastasis of tumors, as well as a poor prognosis by mediating tumor-associated inflammation, promoting cell migration, and facilitating angiogenesis [15,19,20,21,22]. GC patients with a high expression of HMGB1 in tissue and high serum concentrations also have a poor prognosis [22,23]. Extracellular HMGB1 may induce micrometastasis via epithelial-mesenchymal transition (EMT) [24] and angiogenesis [25] by activating the receptor for advanced glycation end products/extracellular signal-regulated kinase/nuclear factor-kappa B (RAGE/Erk/NF-κB) pathway in the early stage of cancer, and it has been identified as the reason for the poor prognosis of patients with high HMGB1 expression levels.

Recombinant human soluble thrombomodulin (rTM) was approved in Japan in 2008 as a novel alternative to heparin for the treatment of disseminated intravascular coagulation syndrome and has contributed to treatment advances [26,27]. Thrombin binds to the EGF-like domain of rTM to form the thrombin-rTM complex, which activates protein C [28]. Activated protein C then inactivates factors VIIIa and Va in the presence of protein S, which exerts anticoagulant effects [29]. Regarding its other functions, rTM binds to HMGB1 via its *N*-terminal lectin-like domain, and assists in the proteolysis of HMGB1 by thrombin, degrading HMGB1 to a less inflammatory form [30,31,32]. The antitumor effects of rTM have also recently been reported [33]. We previously demonstrated that rTM may exert antitumor effects on esophageal squamous cell carcinoma by suppressing the extracellular HMGB1 [21]. 

Therefore, the aim of the present study was to investigate the molecular functions of extracellular HMGB1 in GC cells and verify the antitumor effects of rTM through its targeting of HMGB1.

## 2. Results

### 2.1. Effects of Extracellular HMGB1 and rTM Treatments on GC Cells

HMGB1 expression levels were higher in the GC cell lines than in the non-cancerous cell lines (Appendix A). In experiments involving the addition of extracellular HMGB1, we used the MKN7 and NUGC4 cell lines, which have lower expression levels of HMGB1 than the other GC cell lines. The addition of extracellular rhHMGB1 (200 ng/mL) promoted cell proliferation, and this effect was attenuated by the addition of rTM (100 ng/mL) (Figure 1a). The addition of extracellular rhHMGB1 (200 ng/mL) also promoted migration and invasion, and these effects were attenuated by the addition of rTM (100 ng/mL) (Figure 1b,c). Extracellular HMGB1 transduces cellular signals by interacting with the RAGE/Erk/NF-κB pathway and contributes to stimulating the cell proliferation and migration abilities of GC cells [13,34,35,36]. Alterations in this pathway by extracellular rhHMGB1 and rTM were examined using Western blotting (WB). HMGB1 expression in GC cells was promoted by extracellular rhHMGB1 but was attenuated by the addition of rTM. Similarly, the addition of extracellular rhHMGB1 promoted the phosphorylation of the Erk and NF-κB proteins, and this was attenuated by the treatment with rTM (Figure 1d and Appendix A).

### 2.2. Effects of rhHMGB1 and rTM on Local Tumor Growth and Tumor Metastasis

We evaluated the effects of rhHMGB1 and rTM on local tumor growth and tumor metastasis using subcutaneous tumor and liver metastasis NUGC4 xenograft models of BALB/c nude mice. Figure 2a shows the actual tumors in the subcutaneous model removed on day 20 after the tumor injection, and the volume and weight of each tumor were indicated. The mean tumor volume and weight were significantly larger in the rhHMGB1-treated group than in the non-treated group (*p* < 0.05). Tumors were slightly smaller in the rhHMGB1+rTM-treated group than in the rhHMGB1-treated group. We also measured changes in tumor volumes over time and found the highest growth rate in the rhHMGB1-treated group (Figure 2b). These results suggest that extracellular HMGB1 promoted local tumor growth, while the administration of rTM suppressed extracellular HMGB1, resulting in anti-tumor effects. Alterations in the RAGE/Erk/NF-κB pathway in tumors resected from mice were examined by WB. The phosphorylation of Erk and NF-κB appeared to be enhanced by the treatment with rhHMGB1 and inhibited by the addition of rTM, similar to the changes observed in the expression of HMGB1. 

In the liver metastasis model, only 1 out of 5 mice in the non-treated group had liver metastasis, in contrast to all 4 mice in the rhHMGB1-treated group. In the rhHMBG1+rTM-treated group, 2 out of 5 mice developed liver metastasis (Figure 3a,b). The presence of liver metastasis was pathologically confirmed by HE staining (Figure 3c). High N/C ratio cells were detected around the tumors, while fibrosis and necrosis were observed in the center. These results suggested that the elevated levels of extracellular HMGB1 promoted liver metastasis, and the rTM treatment attenuated these increases, resulting in inhibitory effects on liver metastasis.

To verify these results, we also performed local tumor growth and tumor metastasis experiments under the same conditions at different times and obtained similar results as above (Appendix A).

### 2.3. Relationships between Plasma HMGB1 Concentrations and Clinicopathological Features

The relative expression of HMGB1 was significantly higher in GC tissue (T) than in paired non-cancerous mucosal tissue (NT) (Figure 4a). The plasma concentration of HMGB1 was significantly higher in GC patients before radical gastrectomy than in healthy volunteers (Figure 4b), and the high pre-operative plasma concentration of HMGB1 significantly decreased after surgery (Figure 4c). The plasma concentration of HMGB1 slightly increased with stage progression, and a significant difference was observed in HMGB1 concentrations between stage I and stage III patients (Figure 4d). A correlation was also noted between the plasma HMGB1 concentrations and HMGB1 expression levels in tumor tissue (Figure 4e).

All 56 GC patients from whom pre-operative plasma samples were collected were divided into the Low and High plasma HMGB1 groups using the median of measurements. The High group was correlated with venous invasion (*p* = 0.03), an advanced pN factor (*p* = 0.03), and advanced pStage (*p* = 0.01) (Table 1). Patients in the High group showed significantly worse 5-year recurrence-free survival (RFS) (44.2% vs. 85.7%, *p* < 0.01) and overall survival (OS) (57.4% vs. 89.2%, *p* = 0.01) (Figure 4f). A univariate analysis revealed that venous invasion, the pN factor, pStage, and plasma HMGB1 concentrations correlated with RFS (Table 2). Furthermore, a multivariate analysis identified a high plasma HMGB1 concentration as an independent prognostic factor of RFS (HR 3.20, *p* =  0.04).

## 3. Discussion

Intracellular and extracellular HMGB1 have both been reported to contribute to carcinogenesis, cancer progression, and metastasis [21,25,37,38]. We herein examined the effects of the down-regulation of intracellular HMGB1 using siRNA on cell functions and found that cell proliferation, migration, and invasion abilities were reduced in GC cells, which is consistent with previous findings [39] (Appendix A). Furthermore, in the present study, we focused on extracellular HMGB1 concentrations and investigated the effects of rTM, which may exert anti-tumor effects by decreasing the HMGB1 concentrations.

The present results indicated that extracellular HMGB1 exerted tumor progressive effects that were attenuated by rTM, resulting in the inhibition of tumor growth. Furthermore, lymph node metastasis and stage progression were associated with a high plasma HMGB1 concentration, which was identified as an independent predictive factor for RFS. Clinical results are consistent with the results obtained on cell proliferation, migration, and invasion in the present study. A high extracellular HMGB1 concentration was previously shown to be closely associated with metastasis by promoting the mobility and invasiveness of cancer cells through the activation of the HMGB1-RAGE signaling pathway [24,40]. These reports indicate that, in addition to RAGE, several types of receptors, such as Toll-like receptors (TLRs), are involved in HMGB1 signaling, which has also been implicated in tumor growth [15,34]. As for TLR2, it has been reported to promote tumor growth in breast cancer and colon cancer [41,42]. However, the TLR-mediated tumor-promoting effects of HMGB1 in gastric cancer have been negative, and the HMB1-RAGE-signaling pathway has been reported as the main tumor-promoting factor [21,25,36,43]. The present results also revealed the increased phosphorylation of Erk and NF-κB and the activation of the RAGE/Erk/NF-κB pathway. Moreover, similar alterations in EMT markers were confirmed by the addition of extracellular rhHMGB1 and rTM (Appendix A). Mesenchymal markers, such as *N*-cadherin and vimentin, were increased by extracellular HMGB1, and these effects were attenuated by the addition of rTM. Based on the results of in vitro assays, extracellular HMGB1 also promoted local tumor growth and tumor metastasis in an in vivo assay, while treatment with rTM attenuated these effects, indicating the potential of rTM as a new therapeutic role for GC targeting extracellular HMGB1.

Intracellular HMGB1 is released extracellularly from immune cells during inflammation or from tumor cells due to cytotoxicity, stress, necrosis, and autophagy [43,44,45]. Inflammation, including tumor-related inflammation, is the main factor contributing to the release of HMGB1; however, the release of HMGB1 from tumor cells themselves may also contribute to the high plasma concentration of HMGB1 in the tumor-bearing state [21]. In the present study, high expression levels of HMGB1 in tumor tissue correlated with high plasma HMGB1 concentrations (Figure 4e), which is consistent with previous findings [44]. Although no correlation was observed between HMGB1 expression in tissues and stages, HMGB1 expression levels in tumor tissue were slightly higher in advanced stages (Figure 3b), similar to plasma HMGB1 concentrations (Figure 4d).

HMGB1 has been reported to play a role in micrometastasis from the early stages of cancer through EMT and angiogenesis [24,25]. In the present study, elevated concentrations of extracellular HMGB1 affected local tumor growth and distant metastasis to the liver. Moreover, high plasma HMGB1 concentrations were correlated with metastatic potential, such as venous invasion and lymph node metastasis, and worse RFS in GC patients. When focusing on therapeutic applications, the suppression of this metastatic potential targeting extracellular HMGB1 may effectively improve the prognosis of GC.

rTM is a drug that is approved in Japan for the treatment of disseminated intravascular coagulation syndrome. It binds to HMGB1 via its *N*-terminal lectin-like domain and assists in the proteolysis of HMGB1 by thrombin [30,31,32]. This molecular mechanism inhibits the RAGE pathway and NF-kB phosphorylation, which have recently been reported to exert anti-tumor effects [21,33]. The administration of rTM decreased inflammatory cytokines and inhibited angiogenesis via the suppression of HMGB1 in non-tumor diseases in gynecology [46,47]. On the other hand, elevated concentrations of extracellular HMGB1 from the early stages of GC activated the RAGE pathway and promoted the release of inflammatory cytokines such as IL-8 and TNFα, leading to metastasis via angiogenesis and EMT [24,25]. The inhibition of the HMGB1/RAGE pathway by rTM in the present study may have inhibited early metastasis through the suppression of inflammatory cytokine release, EMT, and angiogenesis. Since rTM is not a cytotoxic agent that is commonly used in the treatment of GC, its inhibitory effects on primary tumors will be weak from a clinical point of view. However, when used in combination with cytotoxic agents, it may suppress the release of extracellular HMGB1 from necrotic cells, inflammatory cells, and tumor cells, and prevent the metastasis of remaining tumor cells.

There are several limitations that need to be considered in the present study. Analyses of plasma HMGB1 concentrations and HMGB1 expression levels in tumor tissue were only performed in a retrospective cohort with a small number of samples. Further studies are needed to elucidate the mechanisms by which rTM attenuates the effects of HMGB1. Limited information is currently available on the functional interaction between intracellular and extracellular HMGB1, and more detailed analyses are required to clarify how they relate to each other in tumorigenesis. In the liver metastasis model assay, the administration of rhHMGB1 promoted metastasis while rTM suppressed this effect. However, it was not possible to confirm HMGB1 levels in metastatic tissues and plasma due to the influence of sample quantity and quality; therefore, further studies are warranted.

In conclusion, plasma HMGB1 concentrations are a prognostic marker in GC patients. Extracellular HMGB1 promotes the progression of GC cells and, at the same time, has the potential to become a novel therapeutic target for rTM.

## 4. Materials and Methods

### 4.1. Cell Lines and Cell Culture

The human GC cell lines, NUGC4 (RCB1939), MKN7 (RCB0999), HGC27 (RCB0500), MKN28 (RCB1000), MKN45 (RCB1001), and MKN74 (RCB1002) and the immortalized fibroblast cell line, WI-38, were obtained from the RIKEN BioResource Research Center Cell Bank (Tsukuba, Japan). The immortalized human mesothelial cell line, Met-5A, was purchased from the American Type Culture Collection (Rockville, MD, USA), and the human umbilical vein endothelial cell line, HUVEC from PromoCell (Heidelberg, Germany).

WI-38, Met-5A, and all GC cell lines besides HGC27 were cultured in Roswell Park Memorial Institute medium (RPMI, Nacalai Tesque, Kyoto, Japan) with 10% fetal bovine serum (FBS, System Biosciences, Palo Alto, CA, USA), 100 µg/mL streptomycin, and 100 U/mL penicillin. HGC27 was maintained in Dulbecco’s Modified Eagle Medium including the same amount of FBS and antibiotics. HUVEC were cultured in endothelial basal medium (Lonza, Allendale, NJ, USA) with the endothelial growth supplement SingleQuots (EGM-2; Lonza). All cells were cultured in a humidified 5% carbon dioxide incubator at 37 °C.

### 4.2. RNA Extraction

Total RNA was extracted from frozen tissue samples using the AllPrep^®^ DNA/RNA/miRNA Universal Kit (Qiagen, Hamburg, Germany) or from cell lines using the miRNeasy^®^ Mini Kit (Qiagen) according to the manufacturer’s instructions. The concentration and quality of all extracted RNA were verified using the NanoDrop 1000 spectrophotometer (Thermo Fisher Scientific, Waltham, USA).

### 4.3. Quantitative Reverse Transcription-Polymerase Chain Reaction (qRT-PCR) Assay

To investigate mRNA expression, a reverse transcription reaction was performed using 250 ng of RNA extracted from tissues or cell lines with the High-Capacity cDNA Reverse Transcription Kit (Applied Biosystems, Foster City, CA, USA). To analyze the expression of HMGB1 mRNA in tissues and cell lines, qRT-PCR was performed using the StepOnePlus PCR system (Applied Biosystem). cDNA samples were applied to qRT-PCR with TaqPath qPCR Master Mix (ROX) (Applied Biosystem) and specific primers. The following conditions were used to perform qRT-PCR: 95 °C for 10 min, 40 cycles at 95 °C for 15 s, and at 60 °C for 1 min. HMGB1 primers used in the present study were purchased from Applied Biosystems (Hs01923466_g1). We used β-actin (Hs01060665_g1, Applied Biosystems) as an internal control for the normalization of mRNA expression. Cycle threshold (*C*_t_) values were calculated using StepOne Software v2.0 (Applied Biosystem), and the results obtained were evaluated using the 2^−ΔΔ*C*t^ method relative to β-actin.

### 4.4. Experimental Reagents and the Down-Regulated Expression of HMGB1

The recombinant human HMGB1 (rhHMGB1) protein was obtained from Sigma-Aldrich (#SRP6265-10UG, St. Louis, MO, USA) and human rTM from Asahi Kasei Pharma (Tokyo, Japan). They were directly added to culture media and used at concentrations of 200 and 100 ng/mL, respectively, in each experiment. When rTM was used with rhHMGB1, it was added 30 min prior to the addition of rhHMGB1. The concentrations of rhHMGB1 and rTM were determined according to a previous report [21].

Regarding the down-regulation of endogenous HMGB1 expression, control siRNA (#D-001810-10-05, GE Healthcare, Buckinghamshire, UK) or siRNA targeting HMGB1 (#L-018981-00, GE Healthcare) was transfected into HGC27 and MKN74 cells on a six-well culture plate at a final concentration of 10 nM using Lipofectamine RNAiMAX (Thermo Fisher Scientific, CA, USA) according to the instructions provided by the manufacturer. The down-regulation of endogenous HMGB1 expression was verified by qRT-PCR and WB (Appendix A).

### 4.5. Proliferation Assays

The number of viable cells treated with extracellular HMGB1 or si-HMGB1 was evaluated using the colorimetric water-soluble tetrazolium salt assay (Cell counting kit-8; Dojindo Laboratories, Kumamoto, Japan). Cell viability was assessed as absorbance values of 0, 24, 48, and 72 h after treatment. The absorbance value of each well was measured at a wavelength of 450 nm.

### 4.6. Migration and Invasion Assays

The BD BioCoat Matrigel^TM^ Invasion Chamber kit (BD Biosciences, NJ, USA) was used to perform migration and invasion assays. Matrigel was coated on the upper surface of the 6.4-mm filter with 8-μm pores for the invasion assay, but not coated for the migration assay. To evaluate the effects of the extracellular HMGB1 and rTM treatments, the migration and invasion abilities of NUGC4 (seeded at 1.0 × 10^5^ cells/well) and MKN7 (1.0 × 10^5^ cells/well) cells were compared among non-treated cells, cells treated with rhHMGB1 (200 ng/mL), and cells pre-treated with rTM (100 ng/mL, 30 min) followed by the rhHMGB1 treatment (200 ng/mL). Cells were seeded into the upper Boyden chamber containing RPMI without FBS, and the lower chamber was filled with RPMI with FBS. After the incubation of cells at 37 °C for 48 h, migratory or infiltrative cells on membranes were fixed and stained with the Diff-Quik stain (Sysmex, Kobe, Japan). The number of nuclei of stained cells was directly counted in six random fields under a microscope, and the mean value was analyzed. The culture medium was not changed after treatment.

In the functional analysis of the down-regulation of endogenous HMGB1 expression, transfected HGC27 cells (0.25 × 10^5^ cells/well) and MKN74 cells (1.0 × 10^5^ cells/well) were also examined using the Transwell migration assay and Matrigel invasion assay as described above.

### 4.7. Enzyme-Linked Immunosorbent Assay (ELISA)

Extracellular HMGB1 protein concentrations in the plasma of patients were measured using HMGB1 ELISA Kit II (Shino-Test Corp., Kanagawa, Japan) according to the instructions provided by the manufacturer. All 56 GC patients included in the present study were divided into two groups, the Low and High HMGB1 groups, based on the median value of measurements, 5.0 ng/mL.

### 4.8. WB Analysis

Anti-HMGB1 (Rabbit, 1:1000; #3935), anti-t-Erk1/2 (Rabbit, 1:1000; #4695), anti-p-Erk1/2 (Rabbit, 1:1000; #9101), anti-t-NF-κB-p65 (Rabbit, 1:1000; #8242), anti-p-NF-κB-p65 (Rabbit, 1:1000; #3033), anti-*N*-Cadherin (Rabbit, 1:1000; #4061), anti-E-Cadherin (Rabbit, 1:1000; #3195), anti-Vimentin (Rabbit, 1:1000; #5741), anti-Snail (Rabbit, 1:1000; #3879), and anti-ZEB1 (Rabbit, 1:1000; #3396) antibodies were purchased from Cell Signaling Technology (Cell Signaling Technology, Danvers, MA, USA). Cells were lysed and protein was extracted using M-PER Mammalian Protein Extraction Reagent (Thermo Fisher Scientific). Protein concentrations were measured using Protein Assay Rapid Kit Wako II (Wako, Tokyo, Japan). Cell lysates with 15 µg of total protein were separated by sodium dodecyl sulfate-polyacrylamide gel electrophoresis and then transferred to polyvinylidene difluoride membranes (GE Healthcare). Membranes were subsequently probed with the indicated antibodies, and protein expression was evaluated using a suitable secondary antibody and the ECL Plus Western Blotting Detection System (GE Healthcare).

### 4.9. Xenograft Nude Mouse Model

The NUGC4 xenograft model of five-week-old female BALB/c nude mice (Shimizu, Kyoto, Japan) was used for all animal experiments under protocols approved by the Animal Ethics Committee of the Kyoto Prefectural University of Medicine (M2020-284).

To assess local tumor growth, a suspension of 5.0 × 10^5^ NUGC4 cells in 50 µL of Matrigel (Corning International, Corning, NY, USA) and 50 µL of phosphate-buffered saline (PBS; Nacalai Tesque) was subcutaneously injected into the left flank of each mouse. Three mice were prepared for the following groups: a non-treated group, rhHMGB1-treated group, and rhHMGB1+rTM-treated group. The treatment protocol is shown in Appendix A. Each drug was percutaneously administered into the peritumor area on days 4, 8, and 12. The non-treated group was administered 100 µL of PBS, the rhHMGB1-treated group with 0.5 µg of rhHMGB1, and the rhHMGB1+rTM-treated group with 0.5 µg of rhHMGB1 and 0.25 µg of rTM. rTM was injected 30 min prior to rhHMGB1. Tumor sizes were measured using a digital caliper every 4 days, and mice were sacrificed 20 days after cell transplantation. Tumor volumes were estimated using the following formula: tumor volume (mm^3^) = width^2^ (mm^2^) × maximum length (mm)/2.

To assess tumor metastasis, a suspension of 1.0 × 10^6^ NUGC4 cells in 50 µL of PBS was injected into the spleen under laparotomy. Five mice were prepared for each group shown above, and the treatment protocol was shown in Appendix A. Each drug was administered on days 1, 3, and 5. The non-treated group was intravenously administered 50 µL of PBS, the rhHMGB1-treated group with 0.5 µg of rhHMGB1, and the rhHMGB1+rTM-treated group with 0.5 µg of rhHMGB1 and 0.25 µg of rTM. rTM was injected 30 min prior to rhHMGB1. Mice were sacrificed 21 days after cell transplantation, and the whole liver of each mouse was resected for evaluation. One mouse in the rhHMGB1-treated group died during the course of treatment, and the remaining 4 mice were used for evaluation.

### 4.10. Patients and Clinical Samples

Pre-operative plasma samples were obtained from 56 GC patients. Patients underwent radical gastrectomy with lymphadenectomy at Kyoto Prefectural University of Medicine Hospital (Kyoto, Japan) between March 2014 and June 2017 and were diagnosed with pathological stages I–III;. Clinicopathological characteristics are shown in Appendix A. Post-operative blood samples were collected from 10 patients at the first outpatient visit after surgery. Plasma samples of 26 healthy volunteers were also collected. Tumor tissues and adjacent normal gastric mucosal tissues were obtained from the resected specimens of 33 GC patients, with plasma samples also being collected from 18 patients. Patients with a previous history of other cancers within the past 5 years were excluded. Clinicopathological findings and postoperative courses were obtained from medical records and databases. The tumor stage was classified according to the 8th edition of the International Union Against Cancer (UICC)/Staging System for Tumors, Nodules, and Metastases [48].

Collected blood samples were immediately processed with a three-spin protocol (i.e., 350 rcf× *g* for 30 min, followed by 700 rcf× *g* for 5 min, and 1600 rcf× *g* for 5 min) to ensure minimal residual cellular components, separated into plasma, and stored at −80 °C. Tumor tissues and adjacent normal gastric mucosal tissues were also stored at −80 °C prior to the extraction of RNA.

### 4.11. Statistical Analysis

All statistical analyses were performed using the statistical software JMP^®^ 13.2.1 (2016 SAS Institute Inc., Cary, NC, USA). The Wilcoxon signed-rank test or Student’s *t*-test was used to compare differences between paired or unpaired samples, and categorical variables in two groups were compared using the chi-squared test. Survival curves for RFS and OS were calculated and plotted using the Kaplan–Meier method and compared using the Log-rank test. The Cox proportional hazard regression model was used to perform a multivariate survival analysis. *p* < 0.05 was considered to indicate a significant difference. All assays were performed in triplicate, except for the in vivo assay.

## Figures and Tables

**Figure 1 ijms-23-03264-f001:**
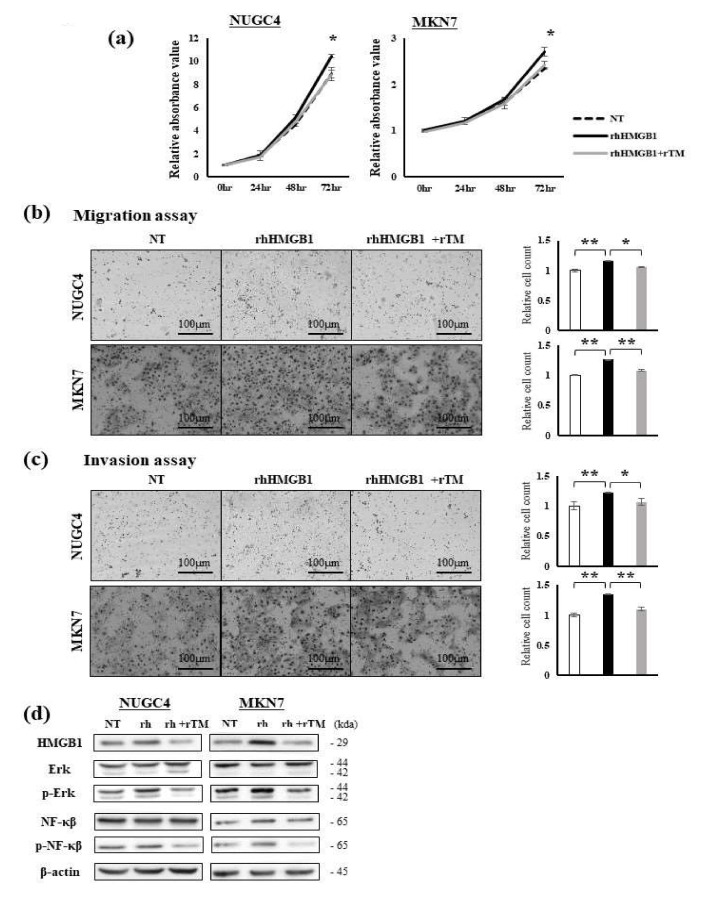
Effects of rhHMGB1 and rTM on GC cells. (**a**) Cell proliferation was evaluated among non-treated NUGC4 and MKN7 cells, cells treated with rhHMGB1 (200 ng/mL), and cells pre-treated with rTM (100 ng/mL, 30 min) followed by rhHMGB1 (200 ng/mL) at the indicated times. Data represent the mean absorbance value ± SD. * *p* < 0.05. (**b**,**c**) The migration (**b**) and invasion (**c**) abilities of NUGC4 and MKN7 cells were evaluated among non-treated cells, cells treated with rhHMGB1 (200 ng/mL), and cells pre-treated with rTM (100 ng/mL, 30 min) followed by rhHMGB1 (200 ng/mL) using the Transwell migration assay and Matrigel invasion assay, respectively. Data represent the mean ± SD of the relative cell count. * *p* < 0.05 and ** *p* < 0.01. (**d**) Alterations in protein expression in the RAGE/Erk/NF-κB pathway by the rhHMGB1 and rTM treatments were evaluated using Western blotting. Whole lysates of NUGC4 and MKN7 cells were obtained under the conditions of no treatment, a treatment with rhHMGB1 (200 ng/mL, 48 h), and a pre-treatment with rTM (100 ng/mL, 30 min) followed by rhHMGB1 (200 ng/mL, 48 h).

**Figure 2 ijms-23-03264-f002:**
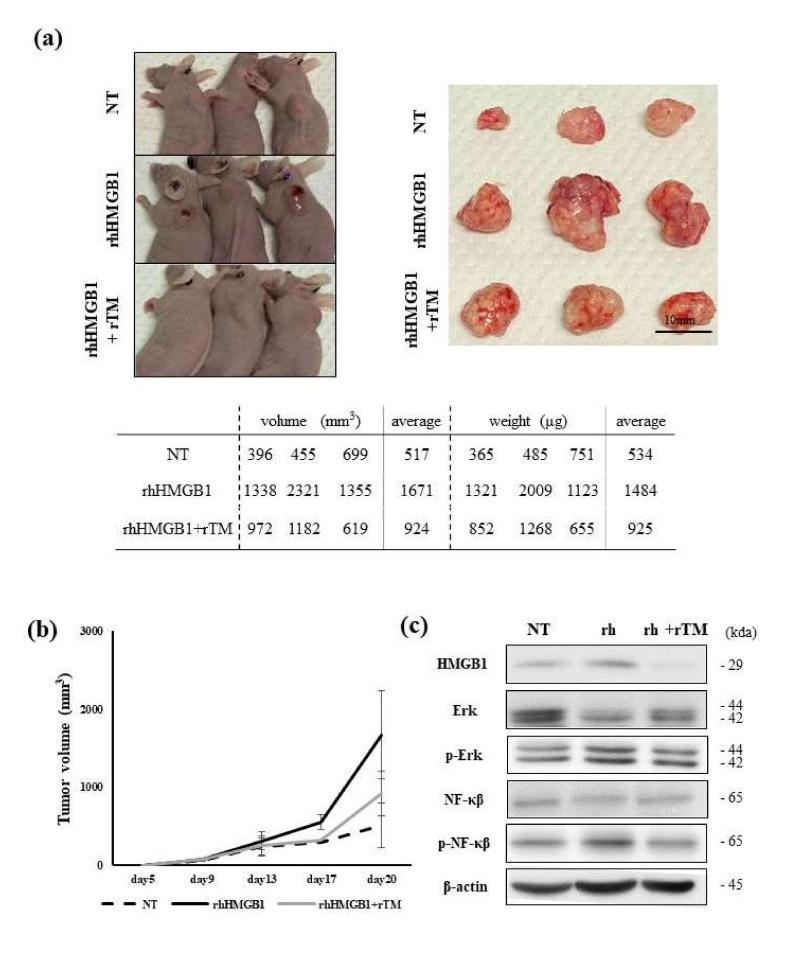
Effects of rhHMGB1 and rTM on local tumor growth in a subcutaneous tumor model. (**a**) Comparisons of subcutaneous tumors among the following three groups: a group treated with only PBS, treated with rhHMGB1, and treated with rhHMGB1 and rTM (*n* = three, respectively). The treatment schedule was the same for all groups (Appendix A). Tumors were resected after sacrifice, and their volume (mm^3^) and weight (µg) were measured. Tumors were slightly larger in the group treated with rhHMGB1 alone. (**b**) Time course of increases in local tumor volumes among the 3 groups. Tumor sizes were measured from the body surface on the indicated days and tumor volumes were calculated. Data represent the mean ± SD of tumors. Alterations in the RAGE/Erk/NF-κB pathway of tumors resected from mice were examined by WB. The phosphorylation of Erk and NF-κB was slightly enhanced by the treatment with rhHMGB1 and inhibited by the addition of rTM. (**c**) Alterations in the RAGE/Erk/NF-κB pathway of tumors resected from mice were examined by WB. The phosphorylation of Erk and NF-κB was slightly enhanced by the treatment with rhHMGB1 and inhibited by the addition of rTM. Representative data from a few experiments are shown.

**Figure 3 ijms-23-03264-f003:**
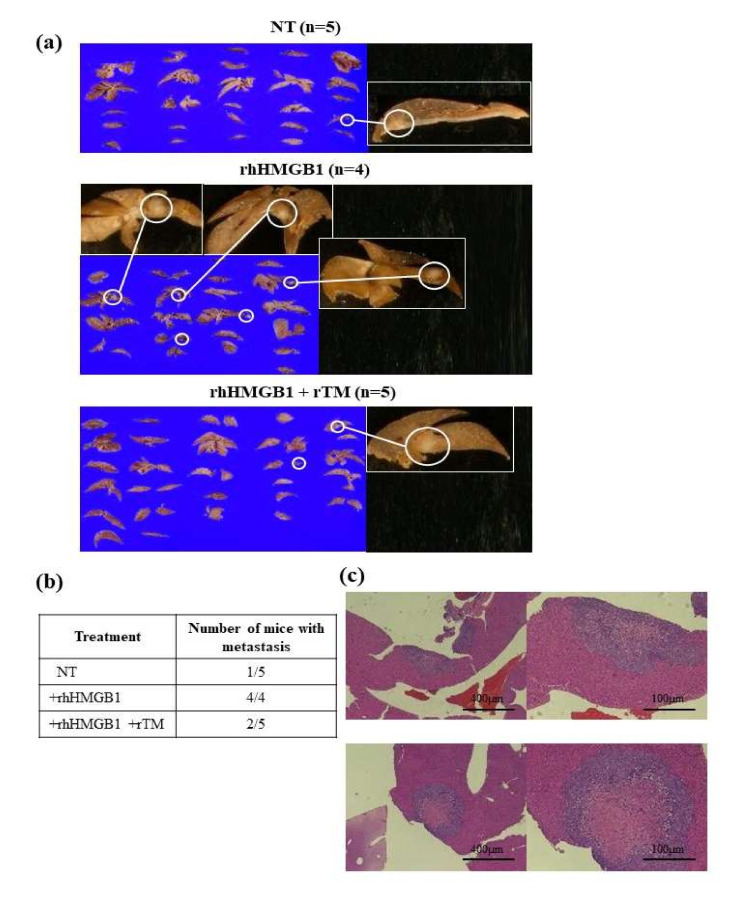
Effects of rhHMGB1 and rTM on tumor metastasis in the liver metastasis model. (**a**) Whole liver samples from all mice used in the experiments were removed and fixed in formalin after sacrifice (*n* = 4 or 5). The treatment schedule was the same for all groups (Appendix A). All cut planes of each liver are shown. The circled area in (**a**) indicates liver metastasis, and an enlarged view is also shown. (**b**) The number of mice with liver metastasis is summarized. All mice in the group treated with rhHMGB1 had liver metastasis, which was more frequent than in the group treated with only PBS, and the addition of rTM reduced liver metastasis. (**c**) Liver metastasis was confirmed by HE staining (at ×40 and ×100 magnification, respectively). High N/C ratio cells were observed around the tumor, and fibrosis and necrosis were noted in the center. Representative data are shown.

**Figure 4 ijms-23-03264-f004:**
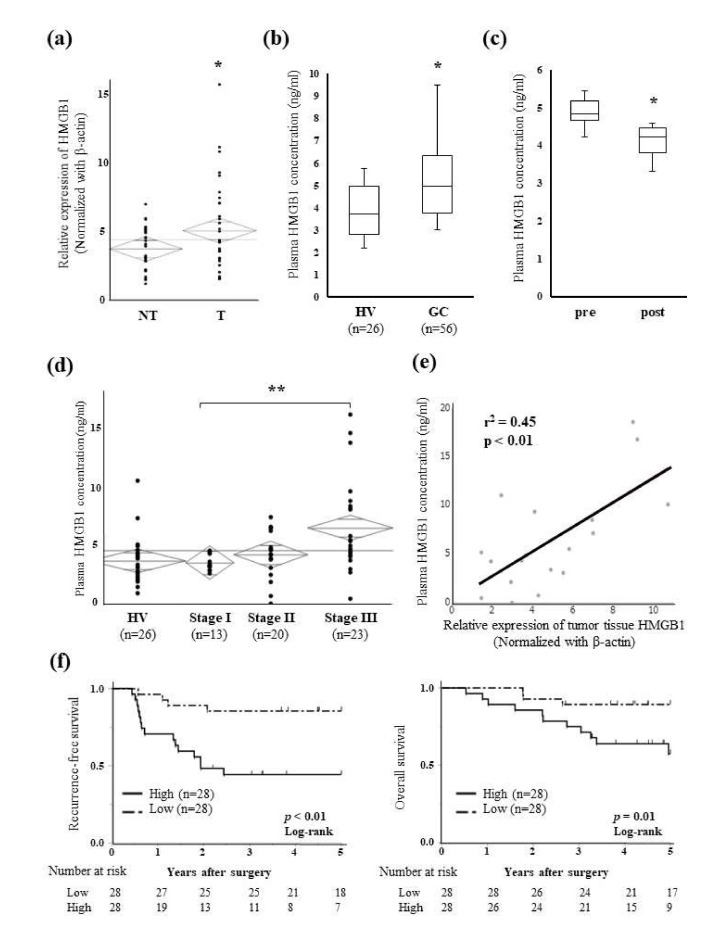
HMGB1 concentrations in tissue and plasma samples and prognostic effects. (**a**) The relative expression of HMGB1 in GC tissue (T) and paired non-cancerous mucosal tissue (NT) was evaluated by qRT-PCR (*n* = 33). Results were analyzed using the paired *t*-test. * *p* < 0.05. (**b**) Plasma HMGB1 concentrations in 56 GC patients and 26 healthy volunteers (HV) were measured by an enzyme-linked immunosorbent assay. Differences were analyzed by the Wilcoxon signed-rank test. * *p* < 0.05. (**c**) Comparison of pre- and post-operative plasma HMGB1 concentrations in 10 GC patients. Plasma HMGB1 concentrations significantly decreased after surgery. Results were analyzed by the Wilcoxon signed-rank test. * *p* < 0.05. (**d**) Plasma HMGB1 concentrations at each stage in GC patients and healthy volunteers (HV). Concentrations slightly increased with stage progression, and a significant difference was observed in concentrations between stage I and III patients (the Mann–Whitney U test). ** *p* < 0.01. (**e**) Relationships between plasma HMGB1 concentrations and HMGB1 expression levels in tumor tissue were examined (*n* = 18). A linear regression model was used for the analysis. (**f**) Kaplan–Meier curves for RFS (left) and OS (right) in GC patients according to pre-operative plasma HMGB1 concentrations. HMGB1 concentrations were divided into two groups, High and Low, by the median value. Differences were analyzed by the Log-rank test.

**Table 1 ijms-23-03264-t001:** Relationships between plasma HMGB1 concentrations and clinicopathological factors in GC patients.

Variables		High (*n* = 28)	Low (*n* = 28)	*p*-Value ^b^
Age (years)	≥70 <70	14 14	11 17	0.41
Sex	Male Female	22 6	16 12	0.08
BMI (kg/m^2^)	<22 ≥22	13 15	16 12	0.42
Location ^a^	U ML	6 22	4 24	0.48
Macroscopic type ^a^	0–2 3–4	13 15	17 11	0.28
Size (mm)	≤50 >50	13 15	15 13	0.59
Differentiation ^a^	Well/moderate Poor	16 12	12 16	0.28
pT factor ^a^	T 1–2 T 3–4	8 20	13 15	0.16
pN factor ^a^	N 0 N 1–3	5 23	12 16	0.03
ly ^a^	0 1–3	8 20	7 21	0.76
v ^a^	0 1–3	5 23	12 16	0.03
pStage ^a^	1–2 3	12 16	21 7	0.01

^a^ According to the 8th edition of the UICC/Staging System for Tumors, Nodules, and Metastases, ^b^ *p*-values are from the Log-rank test, BMI; body mass index, U; upper, M; middle, L; lower.

**Table 2 ijms-23-03264-t002:** Recurrence-free survival analysis.

Variables		*n* = 56	Univariate		Multivariate Analysis		
			5-yr RFS (%)	*p*-Value ^b^	Hazard Ratio	95% CI	*p* ^b^
Age (years)	≥70 <70	25 31	58.5 70.9	0.22			
Sex	Male Female	38 18	64.8 66.6	0.83			
BMI (kg/m^2^)	<22 ≥22	29 27	67.7 62.9	0.68			
Location ^a^	U ML	10 46	56.2 67.3	0.29			
Macroscopic type ^a^	0–2 3–4	30 26	69.0 61.3	0.57			
Size (mm)	≤50 >50	28 28	62.9 67.8	0.78			
Differentiation ^a^	Well/moderate Poor	28 28	63.0 67.6	0.70			
pT factor ^a^	T 1–2 T 3–4	21 35	76.1 58.7	0.18			
pN factor ^a^	N 0 N 1–3	17 39	94.1 52.5	<0.01	-		
ly ^a^	0 1–3	15 41	71.4 63.3	0.55			
v ^a^	0 1–3	17 39	88.2 55.2	0.02	ref 2.41	0.57–16.3	0.24
pStage ^a^	1–2 3	33 23	81.2 43.0	<0.01	ref 2.08	0.74–6.55	0.16
Plasma HMGB1 concentration	High Low	28 28	44.2 85.7	<0.01	3.20 ref	1.03–12.0 -	0.04

^a^ According to the 8th edition of the UICC/Staging System for Tumors, Nodules, and Metastases, ^b^ *p*-values are from the Log-rank test, RFS; recurrence-free survival, BMI; body mass index, U; upper, M; middle, L; lower.

## Data Availability

The datasets used or analyzed during the present study are available from the corresponding author upon reasonable request.

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
