# Peer review of "Role of Extracellular High-Mobility Group Box-1 as a Therapeutic Target of Gastric Cancer"

_ijms, 2022, doi:10.3390/ijms23063264_

Round 1
Reviewer 1 Report
In this manuscript, the author investigates the molecular functions of extracellular HMGB1 in GC cells and verify the antitumor effects of rTM through its targeting of HMGB1. GC proliferation, migration, and invasion abilities were promoted by increases in extracellular HMGB1 concentrations and alleviated by rTM. In my opinion, this manuscript deserves to be recommended for publication in International Journal of Molecular Sciences. However, there are some major or minor issues need to be address before this article to be published.
- How dose rTM inhibit the HMGB1/RAGE pathway? The inflammatory cytokines test and angiogenesis assay are needed.
- The figure s3 should be moved to manuscript because it is a necessary data to support your result. Since the belt of the western blot is not clearly, the author can repeat it once more or add immunofluorescence analysis for EMT markers to confirm the western blotting results.
- In figure 1b, figure 1c, figure 2a and figure 3c, please add the scale bar clearly.
Reviewer 2 Report
In their study including mouse xenograft models and human patient tissues, Takaki and coworkers address the interplay of the danger-associated molecular pattern HMBG1 and thrombomodulin in gastric cancer. This is a highly relevant and important topic and it is of interest that HMGB1 is found increased in gastric cancer. In particular the protective effect of recombinant human thrombomodulin against tumor growth is an interesting finding. The performed analyses appear technically sound. I have a number of comments, which the authors need to address when revising their manuscript.
- In the introduction, the authors should briefly introduce the microbiota’s influence on protease-activated receptor signaling and its role in gastrointestinal malignancy (Reinhardt C, Nature, 2012; Pontarollo G, Cancers, 2019; Wojtukiewicz MZ, Cancers, 2019). Please also discuss factors that are causative or predispose to gastric cancer.
- The role of HMGB1 as a Toll-like receptor-activating molecule needs to be discussed. Please also address the expression of Toll-like receptors in the GC cell lines relative to non-cancerous cell lines as this is an additional factor that promotes gastrointestinal cancers (Scheeren FA, Nat Cell Biol., 2014). Was the enhanced cell proliferation that was induced by HMGB1 treatment mediated via TLR2 signaling (Hörmann N, PLoS One, 2014)? siRNA silencing or inhibitor experiments need to be performed to test this.
- The number of biological replicates in the xenograft mouse tumor model is very low. Please add additional experiments for reliable statistical analyses.
- Although the language is ok, there are numerous hyphen errors that need to be corrected (e.g. in the abstract line 18, line 20 etc.).
- In general, the relative molecular weight of the analysed targets should be indicated for Western blots (e.g. Fig. 1d, Fig. 2c).
Reviewer 3 Report
Role of extracellular high-mobility group box-1 as a therapeutic target of gastric cancer
I appreciate the authors for the efforts for this manuscript. I felt the study design is novel. However, there are some changes needed to improve the quality of the manuscript.
Abstract: Can be improved by framing in a systematic way: Like Hypothesis, Methods, Results and conclusion
Introduction: Can be improved by elaborating the molecular functions of
extracellular HMGB1 in GC cells and verify the antitumor effects of rTM through its targeting of HMGB1
Role of HMGB1 has been reported for sepsis and Cancer need to be differentiated by mentioning the possible mechanism in brief in few lines
Results : Effects of extracellular HMGB1 and rTM treatments on GC cells.
Authors mentioned about the The addition of extracellular rhHMGB1 (200 ng/ml) promoted cell proliferation, and this effect was attenuated by the addition of rTM (100 ng/ml). What is the reason to select the specific dose of rhHMGB1 and rTM?
Effects of rhHMGB1 and rTM on local tumor growth and tumor metastasis: For this study authors only mentioned the picture with 3 animals per group: It would be suggestible to keep at least 6 animals per group to get reliable results with low variability. Please justify this.
For Fig 1; Authors provided the results image of western blotting, it would be great to provide the Raw blots as supplementary files.
In supplementary fig 1; Relative expression of HMGB1: The beta actin was not uniform and there is a variation in the blot. Please keep appropriate image of the blot
Supplementary Fig 3: for MKN7 expression, Beta actin was not uniform and visible please keep appropriate image of the blot
Authors concentrated on only targeting extracellular HMGB1, what are the possible effects of intracellular HMGB1 on relapse of GC in a specific regulatory mechanism using rTM.
Round 2
Reviewer 1 Report
This manuscript deserves to be recommended for publication in International Journal of Molecular Sciences.
Author Response
Dear Reviewer,
Thank you for your sincere replies, and they were very informative for us.
We respond to the comments as follows and revise the manuscript accordingly.
Reviewer: 1
Comment 1. How does rTM inhibit the HMGB1/RAGE pathway? The inflammatory cytokines test and angiogenesis assay are needed.
Our response: Thank you for your kind suggestion. As you show, the inflammatory cytokines or angiogenesis will partially effect to HMGB1 levels. However, at first, rTM directly binds to HMGB1 via its N-terminal lectin-like domain and also assists in the proteolysis of HMGB1 by thrombin, degrading HMGB1 to a less inflammatory form (described in p2. lines 93-95). HMGB1/ RAGE pathway is relatively inhibited by decreasing the concentration of extracellular HMGB1. Consequent to this inhibition, the inflammatory cytokines and angiogenesis will be suppressed particularly in clinical and vivo assay. The inflammatory cytokines and angiogenesis may change, but this is the result of the HMGB1 alteration, not the cause. Furthermore, we do not think these downstream changes are related to the main subject of the present study. Therefore, although the topic about inflammatory cytokines test and angiogenesis assay may be a little off, we added some contents about this pathway in Introduction section.
Comment 2. The figure s3 should be moved to manuscript because it is a necessary data to support your result. Since the belt of the western blot is not clearly, the author can repeat it once more or add immunofluorescence analysis for EMT markers to confirm the western blotting results.
Our response: We appreciate for pointing out. We cultured NUGC4 and MKN7 cells once again with addition of rhHMGB1 and rTM and extracted the proteins for Western blotting. We show a new version of result of western blot and there is no change in the results, but some belts are still not clear as you pointed out. Considering the previous reports, we think it will be due to the sensitivity of the antibodies and the limit of their function. Furthermore, the change of EMT markers by rhHMGB1 in GC cells has been well reported so far (Chung HW, Cancer Sci., 2017), and these intracellular changes deviate from the gist of the present study. Therefore, we would like to put the results on the supplement figure as originally planned. We have updated the images on figure s3.
Comment 3. In figure 1b, figure 1c, figure 2a and figure 3c, please add the scale bar clearly.
Our response: Thank you for your accurate suggestions. We have added appropriate scale bars for the figures.

Reviewer 2 Report
In their study including mouse xenograft models and human patient tissues, Takaki and coworkers address the interplay of the danger-associated molecular pattern HMBG1 and thrombomodulin in gastric cancer. This is a highly relevant and important topic and it is of interest that HMGB1 is found increased in gastric cancer. In particular the protective effect of recombinant human thrombomodulin against tumor growth is an interesting finding. The performed analyses appear technically sound. I have a number of comments, which the authors need to address when revising their manuscript.
- In the introduction, the authors should briefly introduce the microbiota’s influence on protease-activated receptor signaling and its role in gastrointestinal malignancy (Reinhardt C, Nature, 2012; Pontarollo G, Cancers, 2019; Wojtukiewicz MZ, Cancers, 2019). Please also discuss factors that are causative or predispose to gastric cancer.
- The role of HMGB1 as a Toll-like receptor-activating molecule needs to be discussed. Please also address the expression of Toll-like receptors in the GC cell lines relative to non-cancerous cell lines as this is an additional factor that promotes gastrointestinal cancers (Scheeren FA, Nat Cell Biol., 2014). Was the enhanced cell proliferation that was induced by HMGB1 treatment mediated via TLR2 signaling (Hörmann N, PLoS One, 2014)? siRNA silencing or inhibitor experiments need to be performed to test this.
- The number of biological replicates in the xenograft mouse tumor model is very low. Please add additional experiments for reliable statistical analyses.
- Although the language is ok, there are numerous hyphen errors that need to be corrected (e.g. in the abstract line 18, line 20 etc.).
- In general, the relative molecular weight of the analysed targets should be indicated for Western blots (e.g. Fig. 1d, Fig. 2c).
Author Response
Dear Reviewer,
Thank you for your sincere replies, and they were very informative for us.
We respond to the comments as follows and revise the manuscript accordingly.
Reviewer: 2
Comment 1. In the introduction, the authors should briefly introduce the microbiota’s influence on protease-activated receptor signaling and its role in gastrointestinal malignancy (Reinhardt C, Nature, 2012; Pontarollo G, Cancers, 2019; Wojtukiewicz MZ, Cancers, 2019). Please also discuss factors that are causative or predispose to gastric cancer.
Our response: Thank you for your kind suggestions. We have briefly introduced the influence and mechanisms of protease-activated receptor signaling on carcinogenesis of gastric cancer in the Introduction section (p2, lines 53-70).
Comment 2. The role of HMGB1 as a Toll-like receptor-activating molecule needs to be discussed. Please also address the expression of Toll-like receptors in the GC cell lines relative to non-cancerous cell lines as this is an additional factor that promotes gastrointestinal cancers (Scheeren FA, Nat Cell Biol., 2014). Was the enhanced cell proliferation that was induced by HMGB1 treatment mediated via TLR2 signaling (Hörmann N, PLoS One, 2014)? siRNA silencing or inhibitor experiments need to be performed to test this.
Our response: We appreciate for pointing out. We have added to Discussion section about the function of HMGB1 as a Toll-like receptor activating molecule (p13, lines 273-279). As you have pointed out, TLR2-signaling is reported to be a downstream pathway of HMGB1 and promote breast and colorectal cancers or other immune and inflammation responses. However, it is also reported that the HMGB1-TLR-signaling is not the main pathway for tumor growth of GC (Zhang QY, Oncology reports, 2015). Therefore, according to the previous studies, we thought the impact of TLR signaling was very small and confirmed the impact of HMGB1-RAGE signaling pathway in the present study. We thought the evaluation of the TLR pathways was important but a little out of the question in the present study.
Comment 3. The number of biological replicates in the xenograft mouse tumor model is very low. Please add additional experiments for reliable statistical analyses.
Our response: Thank you for your comments. Unfortunately, we were not able to conduct new in vivo assays within the deadline of this revision manuscript. However, we had conducted the same assays before in a small scale and the results were similar to those of we showed in the first manuscript. We will add the results of another in vivo assay in the supplement Figure 6 for evaluating the effects of rhHMGB1 and rTM on local tumor growth and tumor metastasis.
Comment 4. Although the language is ok, there are numerous hyphen errors that need to be corrected (e.g. in the abstract line 18, line 20 etc.).
Our response: We appreciate for pointing out. We have corrected our manuscripts for the hyphen errors.
Comment 5. In general, the relative molecular weight of the analysed targets should be indicated for Western blots (e.g. Fig. 1d, Fig. 2c).
Our response: We have added information about the relative molecular weight of the analysed targets for Western Blots.
Round 3
Reviewer 2 Report
After reading the revised manuscript, my recommendation is to accept this work for publication in the International Journal of Molecular Sciences. The authors thoroughly addressed my previous comments.